# Comprehensive Effect of Carbon Tetrachloride and Reversal of Gandankang Formula in Mice Liver: Involved in Oxidative Stress, Excessive Inflammation, and Intestinal Microflora

**DOI:** 10.3390/antiox11112234

**Published:** 2022-11-12

**Authors:** Yuanyuan Wei, Huiru Wang, Yannan Zhang, Jinhua Gu, Xiuying Zhang, Xuhao Gong, Zhihui Hao

**Affiliations:** 1Department of Clinical Veterinary Medicine, College of Veterinary Medicine, China Agricultural University, Beijing 100193, China; 2Key Biology Laboratory of Chinese Veterinary Medicine, Ministry of Agriculture and Rural Affairs, Beijing 100193, China; 3China Institute of Veterinary Drug Control, Beijing 100081, China

**Keywords:** chronic liver injury, nuclear factor erythroid 2-related factor 2, intestinal flora, component analysis

## Abstract

Aim: To systematically evaluate the effect of Gandankang (GDK) aqueous extract in alleviating acute and chronic liver injury. Forty-one chemical compounds were identified by ultra-high performance liquid chromatography-linear trap quadrupole-orbitrap-tandem mass spectrometry (UHPLC-LTQ-Orbitrap-MS) from GDK. All dosages of GDK and Biphenyl diester (BD) improved CCl_4_-induced acute and chronic liver injury. GDK curbed liver fibrosis and blocked the NF-κB pathway to effectively inhibit the hepatic inflammatory response. Additionally, GDK treatment reduced the abundance of *Phascolarctobacterium*, *Turicibacter*, *Clostridium_xlva*, *Atoprostipes*, and *Eubacterium*, in comparison with those in the CCl_4_ mice and elevated the abundance of *Megamonas* and *Clostridium_IV* as evident from 16S rDNA sequencing. Correlation analysis showed that the abundance of *Eubacterium* and *Phascolarctobacterium* was positively correlated with inflammation, fibrosis, and oxidation indexes. This indicates that GDK ameliorates chronic liver injury by mitigating fibrosis and inflammation. Nrf2 pathway is the key target of GDK in inhibiting liver inflammation and ferroptosis. *Eubacterium* and *Phascolarctobacterium* played a vital role in attenuating liver fibrosis.

## 1. Introduction

Chronic liver injuries are highly prevalent worldwide, and they severely affect the daily life of individuals. According to the World Health Organization report released on 23 April 2022, as of 21 April 2022, 11 European countries, including the UK, Spain, Israel, France, and the United States, reported at least 169 cases of acute hepatitis of unknown origin in children [1]. Subsequently, many cases of acute liver injury of unknown etiology have been reported in Asia. Liver fibrosis, which occurs in many liver diseases, is critical in the progression of liver injury to liver cirrhosis and hepatocellular carcinoma [2,3]. Multiple cell types, including hepatocytes, Kupffer cells, and stellate cells, play essential roles in the pathogenesis of liver fibrosis [2]. Damaged hepatocytes are the “initiators” of liver fibrosis, which trigger the release of excessive reactive oxygen species (ROS) and chemotactic cytokines to continuously activate the cascade of inflammation and fibrosis [4]. Despite large investments in new therapies by pharmaceutical industries, no therapy targeting liver fibrosis has been approved yet [5]. Unraveling the mechanism of liver fibrosis and devising safe and effective treatment strategies is, therefore, of high value.

Oxidative stress, mainly due to abundant ROS production and weakening of the antioxidant capacity, is important in the progression of liver fibrosis [6]. Nuclear factor erythroid 2-related factor 2 (Nrf2) is a key factor in regulating the oxidative stress response [7,8]. It binds to the cytoplasmic protein Kelch-like ECH-associated protein 1 (Keap1) under normal conditions [9]. In the presence of excessive free radicals, Keap1 detaches from Nrf2, and free cytoplasmic Nrf2 is transferred to the nucleus, where it activates the genes encoding antioxidant enzymes, such as heme oxygenase (HO-1), NAD(P)H: Quinone oxidoreductase 1 (NQO1), and glutamate-cysteine ligase catalytic subunit (GCLC) [7,10]. In the liver fibrosis model induced by chemical toxins, such as carbon tetrachloride (CCl4) [11], thioacetyl [12], and acetaminophen [13], Nrf2 activation effectively protects against liver fibrosis. Contrarily, deletion of Nrf2 leads to chemically induced disorders of cellular redox homeostasis and xenobiotic metabolism in the liver, resulting in excessive accumulation of ROS, irreversible oxidative damage to DNA, and liver fibrosis [13,14,15,16]. Xu et al. [17] found that in Nrf2-knockout mice, liver injury and fibrosis induced by CCl4 were more serious than in wild-type mice. Similarly, genetic inhibition of Nrf2 worsened inflammation and spontaneous liver fibers in a mouse model of hereditary hemochromatosis [18]. Although the specific role of Nrf2 in the progression of liver fibrosis has been studied, its exact function in complex pathophysiological processes is unclear.

Inflammatory responses also cause liver fibrosis [19]. Persistent liver injury leads to inflammatory pathology and infiltration of inflammatory cells [20]. A key factor in hepatocyte-driven liver fibrosis is the activation of key proinflammatory cytokines regulated by the NF-κB signaling pathway in hepatocytes [21]. The activation of NF-κB in injured hepatocytes causes secretion of various proinflammatory cytokines and chemokines, including interleukin-1 β (IL-1β), tumor necrosis factor-α (TNF-α), IL-6, and monocyte chemoattractant protein-1 (MCP-1) [22]. The NF-κB signaling pathway also plays an important role in the activation of hepatic stellate cell fibrosis [23]. Although activated NF-κB is involved in the development of fibrosis, its exact contribution remains unknown. 

The intestinal microbiota is also related to liver injury and fibrosis in patients with chronic hepatotoxicity [24,25]. The degree of liver fibrosis is closely associated with the bacterial composition of feces in patients with non-alcoholic fatty liver disease. There is sufficient evidence that metagenomic characteristics of the intestinal microbiome indicate the level of cirrhosis [26]. Recent studies show that members of Lactobacillus in the intestine can activate the Nrf2 signaling pathway, prompting protection against liver injury and reducing liver fibrosis [27].

Many herbal remedies used in traditional Chinese medicine have antifibrotic effects [28]. Gandankang (GDK) granule emerged from the traditional prescribing of Yinchenhao, a decoction approved by the Chinese National Medical Products Administration for clinically treating chronic hepatitis and pancreatitis in China [29,30,31,32,33]. Yinchenhao decoction, used in liver fibrosis treatment, was reported to regulate multiple targets, especially those affecting apoptosis-related signaling pathways [34]. Our preliminary research reported that GDK inhibits oxidative stress and inflammation in liver to alleviate liver injury in mice. However, the efficacy of GDK on hepatic fibrosis and the underlying mechanism remain unclear.

In this study, we report that GDK effectively inhibits CCl4-induced liver fibrosis by inhibiting inflammation and oxidative stress. Mechanistically, the attenuation of liver fibrosis by GDK largely depends on Nrf2 activation and subsequent inhibition of the NF-kB pathway. We also found that the intestinal flora plays an important role in the alleviation of liver fibrosis by GDK. We propose that GDK granules may be a promising drug for the treatment of liver fibrosis.

## 2. Materials and Methods

### 2.1. LC-MS Conditions and Chemical Composition Analysis

GDK formula (consisting of 6 herbs: Artemisiae Scopariae Herba, Scutellariae Barbatae Herba, Schisandrae Chinensis Fructus, Gardeniae Fructus, licorice, and Isatidis Radix) was purchased from China Beijing Tongrentang Co., Ltd. A Waters AQUITYUPLCHSST3C18 (2.1 × 100 mm, 1.8 μm) column was used with the mobile phase of (A) 0.1% formic acid water and (B) acetonitrile. The gradient elution is 0~7 min (5–15% B), 7~14 min (15–20% B), 14~21 min (20–23% B), 21~28 min (23–40% B), 28~35 min (40–70% B), 35~42 min (70–95% B), 42~50 min (95% B), and 50~58 min (5% B), respectively. The flow rate was 0.25 mL/min, the column temperature was 35 °C, and the injection volume was 3 μL. In the negative ion mode, the best operating parameters were: Sheath gas flow 30 arb, auxiliary gas flow 10 arb, capillary voltage −35V, electrospray voltage 3.0 kV, tube lens voltage −110V, and capillary temperature 300 °C.

### 2.2. Establishment of Chronic Liver Injury Models

Eight-week-old Institute of Cancer Research (ICR) mice of Specific Pathogen Free (SPF) grade were purchased from Speyford (Beijing, China) Experimental Animal Science and Technology Co., Ltd. (license SCXK (Beijing, China) 2019–0010). Mice were separated into six groups (n = 8). Mice in the control group were perfused with 0.2 mL distilled water for four weeks and treated with corn oil (10 µL/g body weight, BW, i.p) on days 1, 3, and 5. The mice in the model group were given equal volumes of distilled water for four weeks and treated with 10% CCl_4_ solution (10 µL/g BW) three times a week [35,36]. In the low, medium, and high dosage GDK groups, the mice were orally administered 1.17, 2.34, and 4.68 mg/kg BW GDK, respectively, for four weeks, followed by 10% CCl_4_ solution on the same day. Biphenyl diester (BD, 200 mg/kg) was used as the positive control in our study, and its treatment protocol was similar to that of GDK. All experiments involving animals were conducted according to the ethical policies of and procedures approved by the Institutional Animal Care and Use Committee of China Agricultural University (No. Aw20801202–2–2).

### 2.3. Measurement of Cytokines

Cytokines IL-1β (Cat. No. E-EL-M0037c), IL-6 (Cat. No. E-EL-M0044c), and TNF-α (Cat. No. E-EL-M3063) in the liver homogenate and serum were measured using kits purchased from Elabscience Biotechnology Co., Ltd. (Wuhan, China).

### 2.4. Hepatic Function Assays

Serum was obtained by centrifuging blood samples at 1500× *g* for 15 min. The levels of alanine aminotransferase (GPT/ALT, Cat. No. C009-2-1), aspartate aminotransferase (GOT/AST, Cat. No. C010-2-1), lactate dehydrogenase (LDH, Cat. No. A020-2-2), total bile acid (TBA, Cat. No. E003-2-1), and total bilirubin (TBIL, Cat. No. C019-1-1) in the serum or liver homogenate were measured using kits purchased from the Nanjing Jiancheng Bioengineering Institute.

### 2.5. Histopathology of Liver

Liver tissues were separated from each sample and fixed in 4% formaldehyde for 1 week at 25 °C and then embedded in paraffin and sliced into 4 μm sections. The sections were stained with Eosinophil Staining Kit (Solarbio, Cat. No. G3632) and Masson’s Trichrome Stain Kit (fast green FCF method; Solarbio, Cat. No. G1343). After sections were mounted and imaged under a light microscope (DS-Ri2, Nikon, Japan), the slices were graded with reference to Ishak scoring rules (Appendix A). The mean density of collagen volume fraction (CVF) and immunohistochemical reactions were measured by Fiji image analysis. Mean density was calculated as integrated optical density (IOD) divided by area.

### 2.6. Detection of Hepatic Fibrosis and Oxidation Parameters

The levels of hydroxyproline (HYP, Cat. No. A030-2-1), γ-glutamyltransferase (γ-GT, Cat. No. C017-2-1), and hyaluronic acid (HA, Cat. No. H141-1-2) were determined using kits with multi-wavelength measurement system (TECAN, Infinite M200Pro Nanoquant). The enzymatic activities of total superoxide dismutase (T-SOD, Cat. No. A001-1), catalase (CAT, Cat. No. A007-1-1), peroxidase (POD, Cat. No. A084-1-1), and glutathione peroxidase (GSH-px, Cat. No. A005-1-1), as well as the levels of malondialdehyde (MDA, Cat. No. A003-1-2) and glutathione (GSH, Cat. No. A006-2-1), were determined using respective kits, as described by the Nanjing Jiancheng Bioengineering Institute.

### 2.7. Western Blot Analysis

The liver tissue samples were lysed with RIPA buffer (containing 1% phenylmethyl-sulfonyl fluoride and 1% protease inhibitor cocktail) for 30 min on ice and then homogenized and centrifuged at 14,000× *g* for 30 min. The supernatant was collected and stored at 4 °C for western blot analysis. Nuclear Protein Extraction Kit was used for subcellular protein extraction, according to the manufacturer’s protocol.

The protein concentration was determined with a BCA kit. The protein samples were separated on 12% SDS-polyacrylamide gels and transferred onto a PVDF membrane (Millipore, Billerica, MA, USA, Cat. No. IPVH00010). The membrane was blocked with 5% BSA in TBST and then incubated with specific primary antibodies (Appendix A) overnight at 4 °C followed by corresponding HRP-conjugated anti-rabbit secondary antibodies (1:5000) for 3 h at 21 °C. ECL Prime Western Blotting Detection Reagent (Cytiva, Marlborough, MA, USA; Shinjuku City, Japan, Cat. No. RPN2232) for visualization of protein bands with an imaging system (Cytiva) was used. The intensity of bands was analyzed using Fiji image analysis [37].

### 2.8. RNA Extraction, Reverse Transcription, and Q-PCR

Total RNA was extracted from the liver tissue using a total RNA extraction kit (Promega, Madison, WI, USA, Cat. No. LS1040). Single-strand cDNA was reverse-transcribed from the extracted RNA using the GoScript™ Reverse Transcription System according to the manufacturer’s instructions. q-PCR was performed on the CFX96 Touch Real-Time PCR Detection System (Bio-Red, Hercules, CA, USA) using SYBR Green reagent (Promega, Cat. No. A6001). The sequences of forward (F-) and reverse (R-) primers are listed in the Appendix A. The relative expression level of each gene was calculated using the 2^−ΔΔCt^ method.

### 2.9. TUNEL Assay 

Liver sections were prepared and dewaxed according to the instructions in the TUNEL kit manual (Boster, Pleasanton, CA, USA, Cat. No. MK1025) and immunohistochemistry (IHC) instructions (Solarbio, Beijing, China, Cat. No. SA0021). Briefly, endogenous peroxidase was quenched with 3% H_2_O_2_, the cell membrane was permeabilized with proteinase K treatment at 37 °C for 15 min, and the sections were washed thrice with TBS for 2 min, blocked with 5% BSA for 30 min, and incubated with biotinylated anti-digoxin antibody at 37 °C for 2 h. Next, the sections were incubated with TDT and DIG-d-UTP (1:100) at 37 °C for 2 h, washed thrice with TBS for 2 min, and then incubated with SABC. Finally, the stain was developed using a DAB system. Cells showing brown-yellow particles in the nuclei were considered apoptotic.

### 2.10. Determination of Fecal Flora Diversity

The method we used referred to our previous methods [38]. In brief, DNA was extracted from feces, and its quality was assessed using spectrophotometry. Polymerase chain reaction (PCR) was conducted using a diluted genomic DNA template, a KAPA HiFi HotStart ReadyMix PCR Kit, and DNA polymerase. The primer sequences for V3-V4 region amplification were as follows: (338F: 5′-ACTCCTACGGGAGGCAGCAG-3′ and 806R: 5′-GGACTACHVGGGTWTCTAAT-3′). The operational taxonomic unit clusters and species classification analyses, α- and β-diversity, were performed on the Sibiocore platform.

### 2.11. Statistical Analysis

All data are expressed as the mean ± SD, and statistical significance was determined using one-way ANOVA followed by Tukey’s multiple comparison procedure. *p* values are specified as follows: *p* < 0.05 (*), (#); *p* < 0.01 (**), (##).

## 3. Results

### 3.1. Chemical Composition Identification in GDK

To determine chemical ingredients in GDK, sample analysis was performed in the negative ion mode. After subtracting blank and removing peak adducts that were generated in source, the compounds were tentatively identified from their exact mass, isotope patterns, and mass fragmentation patterns. A total of 41 communal compounds were preliminarily identified by LC-Orbitrap-ESI-MS (Table 1, Figure 1). They contained various chemical compositions, such as Sebacic Acid, licoagrochalcone A, Gancaonin C, neoliquiritin, Rutin, Isovitexin, Gancaonin O, Quercetin, Gancaonin R, Chrysophanol, Isorhamnetin, Glycyrrhizic Acid, etc. (Table 1).

### 3.2. GDK Improves Hepatic Function and Structure in Chronic Liver Injury

Subsequently, we explored the effects of GDK on liver structure and function in chronic liver injury induced by 10% CCl_4_. After 10% CCl_4_ treatment, the liver became brittle and white, and the surface of the capsule was rough. BD and different doses of GDK considerably improved the appearance of liver fibrosis. The disordered arrangement of hepatocytes in the pseudolobule included steatosis, necrosis and regeneration, and deviation or absence of the central vein, lymphocytes, monocytes, and newborn bile ducts in the fibrous tissue around the pseudolobule. Furthermore, BD and GDK improved hepatic LFS broadening, pseudolobule formation, cellular lipid degeneration, and eosinophil infiltration induced by 10% CCl_4_ and decreased liver pathological scores (Figure 2A,B).

We analyzed the biochemical factors in serum to evaluate the status of liver function. The levels of serum AST, ALT, TBA, TBIL, γ-GT, and LDH were markedly increased by 10% CCl_4_. Administration of BD and different doses of GDK markedly decreased the levels of serum AST, ALT, LDH, TBA, and γ-GT. There was no significant difference in serum AST and ALT levels between 2.34, 4.68 mg/kg GDK, and the blank groups. CCl_4_ (10%) caused a considerable decrease in serum TP levels, whereas BD and different doses of GDK considerably increased its levels (Figure 2C).

### 3.3. GDK Reduces Inflammation in CCl4-Induced Chronic Liver Injury

Because inflammation often accompanies liver disease, we detected the state of inflammation in chronic liver injury. First, the levels of serum inflammatory cytokines were detected. CCl_4_ (10%) remarkably increased the IL-6, IL-1β, and TNF-α levels. BD and different doses of GDK remarkably decreased the levels of IL-1β and TNF-α, and 2.34 mg/kg BD and 4.68 mg/kg GDK markedly decreased the IL-6 levels (Figure 3A). Additionally, 10% CCl4 caused apoptosis in the liver tissue, and different doses of GDK markedly inhibited liver apoptosis (Figure 3B,C).

In terms of proteins, 10% CCl_4_ markedly promoted the translocation of NF-κBp65 into the nucleus. BD and different doses of GDK considerably inhibited the translocation of NF-κBp65 into the nucleus (Figure 3D).

### 3.4. GDK Suppresses Fibrosis in CCl4-Induced Chronic Liver Injury

Fibrosis is a prominent feature of chronic liver injury, so we detected the fibrotic phenotype in the liver. Masson’s staining showed that intermittent treatment with 10% CCl_4_ induced fibrous tissue hyperplasia and pseudobular formation in the liver and increased the fibrosis score extensively. BD and 1.17 mg/kg GDK had a certain alleviating effect on fibrous tissue proliferation, but a trend of pseudolobular formation was still observed. GDK (2.34 and 4.68 mg/kg) considerably reduced the intrahepatic fibrous tissue proliferation and fibrosis scores and inhibited pseudolobular formation (Figure 4A). Predictably, the abnormal increase in liver tissue HYP levels and serum HA levels was markedly reversed by BD and different doses of GDK (Figure 4B).

### 3.5. GDK Weakens Ferroptosis through the Antioxidant System in CCl_4_-Induced Chronic Liver Injury

Considering the above results, we hypothesized that oxidative stress and ferroptosis play key roles in chronic liver injury. To test this, we first detected the serum and tissue levels of oxidative products and antioxidants. BD and different doses of GDK considerably reduced MDA content in the serum and liver tissue (Figure 5A). Intriguingly, there was no significant difference in CAT levels in the liver tissue among the treatment groups. However, the activity of serum CAT in the CCl_4_ group was highest and was considerably reduced after the treatment. The GSH-Px activity in serum and liver tissue was markedly reduced by CCl_4_, whereas 1.17 and 2.34 mg/kg GDK markedly increased it. Interestingly, different trends were observed for the GST activity in tissue and serum. The activity in serum was considerably increased in the CCl4 group but considerably reduced in the liver tissue. BD and different doses of GDK markedly restored the abnormal expression of GST. In addition, the activity of POD in the liver tissue was also markedly increased in the CCl_4_, 2.34 mg/kg, and 4.68 mg/kg GDK groups. BD and 2.34 mg/kg GDK considerably increased the activity of serum T-SOD, and 2.34 and 4.68 mg/kg GDK considerably increased the content of serum GSH. No significant effect on serum T-AOC was observed in the administration group (Figure 5A).

We further explored the expression of key targets in oxidative stress and the ferroptosis signaling pathway. GDK (2.34 and 4.68 mg/kg) considerably promoted the expression of *Nfe2l2*, *Slc7a11*, and *Gpx4*, while down-regulated the expression of *Keap-1*, *Hmox-1*, *Nqo-1*, and *Acsl4* (Figure 5B).

### 3.6. GDK Restores Intestinal Flora Diversity to Curb Chronic Liver Injury

Compound traditional Chinese medicine often affects the changes in intestinal flora, possibly affecting disease development. However, pertaining to its complexity, the regulation of the intestinal flora has not been revealed. We explored the effect of GDK on the abundance of intestinal flora to further explore the mechanism. First, the species accumulation curve was used to determine the sample size for the detection of diversity (Figure 6A). The Simpson index represents the probability (0–1) that two randomly selected sequences belong to the same classification (species level). The closer the value is to 1, the more uneven the abundance distribution of ZOTUs, showing a more uniform abundance distribution of flora in each group. Analysis of similarity (ANOSIM) was used to investigate whether there were significant differences in community structures between and within groups. The results of ANOSIM based on the Bray–Curtis distance showed marked differences among groups. CCl4 (10%) considerably decreased the abundance of intestinal microflora in mice, which was restored by 2.34 mg/kg GDK (Figure 5B). Euclidean Principal Coordinate Analysis (PCoA) and infinite multi-dimensional calibration (NMDS, Non-Metric Multi-Dimensional Scaling) showed that samples in the 10% CCl_4_, 2.34 mg/kg GDK, and control groups were closely gathered together, and the coordinate positions of the 2.34 mg/kg GDK and control groups were closer, indicating that their species composition was more similar (Figure 6C). Species diversity analysis showed that CCl_4_ and GDK treatment had notable effects on the composition of microflora at genus and phylum levels (Figure 6D,E).

To study the similarity between different samples, we used the unweighted pair-group method with arithmetic mean (UPGMA) cluster analysis to construct a cluster tree of the samples. The distance matrix and UPGMA cluster analyses were performed, and the results were combined with the relative abundance of some species of each sample at the gate level (phylum). The samples in the control and 2.34 mg/kg GDK groups were clustered together, and those in the 10% CCl_4_ group were clustered together. GDK (2.34 mg/kg) restored the change in flora structure caused by CCl_4_, and the regulation of *Firmicutes*, *Bacteroidetes*, *Proteobacteria*, *Romboutsia*, *Acidobacteria*, *Campilobacterota*, etc., was obvious (Figure 6F,G). We analyzed the differences in considerably changed microflora before and after GDK treatment and found that, compared with the model, 2.34 mg/kg GDK markedly downregulated the abundance of *Phascolarctobacterium*, *Clostridium_IV*, and *Megamonas* and considerably upregulated the abundance of *Eubacterium*, *Atopostipes*, *Clostridium clusterXlVa*, and *Turicibacter*. The changes in microflora after GDK treatment tended toward the control group (Figure 7A,B). Furthermore, *Eubacterium* was positively correlated with IL-1β, TNF-α, HYP, HA, and MDA, as well as negatively correlated with GSH-Px. *Phascolarctobacterium* were positively correlated with IL-1β and TNF-α (Figure 7C).

## 4. Discussion

Chronic liver injury mainly refers to hepatocyte necrosis and abnormal liver function owing to various reasons. Their incidence is increasing, and they are among the most serious risk factors for metabolic complications and secondary death, posing a heavy burden on the economy and healthcare system [1]. There is no effective treatment for chronic liver injury [39]. Chinese herbal prescriptions have been used for thousands of years to treat liver injury diseases. GDK is derived from the famous prescription “Yinchenhao decoction” for jaundice treatment by Zhongjing Zhang, a medical saint of the Eastern Han Dynasty, and is named Yinshanlian prescription in the Chinese Pharmacopoeia. We previously found that the protective effect of this prescription on CCl4-induced liver injury was mainly through antioxidation, anti-inflammation, and fibrosis inhibition through p38 MAPK and NF-κB signal transduction pathways. In this study, we established toxin-mediated experimental chronic liver injury models. Different doses of GDK reduced the balloon degeneration of hepatocytes and the levels of oxidative stress and inflammation in liver and serum and protected against liver injury, as indicated by reduced levels of serum AST, ALT, γ-GT, TBA, and TBIL. Through comprehensive application of 16S rDNA sequencing analysis, we systematically evaluated the specific effects of GDK on the intestinal flora and the expression levels of proteins related to oxidative stress and ferroptosis. 

Ferroptosis plays a crucial regulatory role in liver inflammation and fibrosis [40]. It is a dynamic process involved in iron metabolism and lipid peroxidation [40,41,42]. Iron overload induces membrane lipid peroxidation during ferroptosis because of the Fenton reaction in multiple subcellular organelles [41]. Oxidative damage of membrane lipids is central to the execution of ferroptosis [43,44]. The final product of lipid oxidation, MDA, not only affects the activities of key enzymes in mitochondria but also aggravates membrane damage. GDK can also be used as an antioxidant to remove excess ROS and reduce the content of MDA in serum and liver tissue in chronic liver injury. The main members of antioxidant system include antioxidant enzymes and nonenzymatic antioxidants [45,46]. We then determined the levels of SOD, GSH-Px, CAT, POD, and GST. GDK increased the levels of SOD, GSH-Px, and POD in serum and liver tissue in chronic liver injury models in keeping with the previous study [47]. Interestingly, Serum CAT activity in the model group was significantly higher than that in the control and treated groups, but there was no significant difference in the tissue levels. The serum activity of GST was also very high in the chronic liver injury model group and was significantly decreased after administration of different doses of GDK. However, at the tissue level, the trend was opposite to that in serum. The peak serum GST level in liver disease appeared earlier and was higher than that of AST. Therefore, the serum GST level is a very sensitive index of hepatocyte injury. GSH, a nonenzymatic antioxidant, is a coenzyme of many enzymes, such as GSH-Px, mainly involved in scavenging ROS and in protecting the body from oxidative damage. GDK (2.34 and 4.68 mg/kg) significantly increased the content of serum GSH in mice with chronic liver injury. 

Nrf2 is a central player in the regulation of antioxidant molecules in cells [48]. Nrf2 activation in patients with non-alcoholic steatohepatitis correlates with the grade of inflammation. Functional analysis in mice demonstrated that Nrf2 activation in chronic liver disease ameliorates fibrogenesis, initiation, and progression of HCC [49]. It controls cellular antioxidant systems in hepatocytes, playing a key role in protecting against intracellular and environmental stress [7]. We found that Nrf2 expression decreased with fibrosis severity, whereas it was high in normal liver tissue. Activation of the Nrf2 signal transduction pathway activates the antioxidant system. Nrf2 is constantly degraded by Keap1 and is activated by the inhibition of Keap1 [50]. The detection of Keap1 also shows that GDK weakens the inhibition of Nrf2, thus promoting its nuclear translocation, and thereby increases the expression of HO-1 and NQO-1. Interestingly, the expression of HO-1 is not completely consistent with that of Nrf2, which may be related to another function of HO-1 in promoting the production of Fe2+. The Keap1–Nrf2 antioxidant signaling pathway is a negative regulator of ferroptosis and determines the sensitivity to ferroptosis inducers such as erastin, sulfasalazine, and sorafenib, which inhibit xCT. SLC7A11 is a key component of system Xc2, an amino acid antiporter that typically mediates the exchange of extracellular L-cystine and intracellular L-glutamate across the plasma membrane. Suppression of SLC7A11 leads to GSH depletion and increased ferroptosis. Interestingly, SLC7A11 is a transcriptional target of Nrf2 [51,52].

Furthermore, we found that GDK inhibited ferroptosis by activating the Nrf2 pathway, which influences liver inflammation and the fibrotic phenotype in mice with liver injury. Ferroptosis can promote liver fibrosis [17]. We showed that not only the keap1–Nrf2 pathway was inhibited, but the classical NF-κB pathway was also activated, and significant cell death occurred during chronic liver injury. In chronic liver injury, the proliferation of fibers was significant, and there was a tendency to form false lobules [53]. Our research shows that HA are also introduced as characteristic indicators of fibrosis [47]. As expected, the increase in serum HA levels, which is an indicator of fibrosis initiation, significantly decreased with all doses of GDK, indicating that the antagonistic relationship was gradually recovered, thereby reducing the inflammatory response and cell death.

Based on our sequencing results for intestinal flora in mice with chronic liver injury, we found that GDK upregulated the abundance of *Turicibacter,* which was then found to interact with bile acid (BA) [54]. However, unlike our results, it is reported that the relative abundance of *Turicibacter* is strongly positively correlated with liver MDA levels [55]. GDK also increases the abundance of Clostridium cluster XlVa, Atopostipes, and *Eubacterium*. *Eubacterium* is associated with bile acid regulation [56]. Indeed, *Eubacterium* spp., along with other genera, such as Roseburia and Clostridium, constitute a major reservoir of BSHs in the gut [57]. Besides, *Eubacterium* spp. contribute to gut and hepatic health through modulation of bile acid metabolism [57]. Clostridium cluster XlVa, Atopostipes, and *Eubacterium* are also related to the synthesis of amino acids and short-chain fatty acids [55,57]. The increase in the abundance of these bacteria may play a pivotal role in the recovery of liver function, but their internal causal relationship needs to be further explored. We also found that GDK decreased the abundance of *Phascolarctobacterium*, Clostridium_IV, and *Megamonas* flora. *Phascolarctobacterium* is related to the production of short-chain fatty acids such as acetate and propionate and has a certain detoxification effect [58]. The role of *Clostridium_IV* and *Megamonas* in liver diseases has not been reported. Contrarily, by comparing with the microflora related to chronic liver fibrosis or liver cirrhosis queried in the Disbiome database, *Phascolarctobacterium* and *Eubacterium* were not only found to be associated with liver cirrhosis but were also verified by our sequencing results. Therefore, we have two propositions to further explore the mechanism in the future. On the one hand, we can select *Clostridium_IV* and *Megamonas*, which have not been reported in the study on chronic liver injury, and perform macro-genome sequencing, followed by their isolation, purification, and animal verification to unravel their effect on chronic liver injury. On the other hand, we can also choose *Phascolarctobacterium* and *Eubacterium*, which were identified by querying the database and make use of operations such as fecal-oral transplantation to further explore the specific causal relationship between *Phascolarctobacterium*, *Eubacterium*, and chronic liver injury.

## 5. Conclusions

Our findings show that different doses of GDK can alleviate chronic liver injury and 2.34 mg/kg GDK has the best comprehensive effect. The effect of GDK on chronic liver injury may be through the activation of the Keap1/Nrf2 pathway and inhibition of ferroptosis, alleviating liver fibrosis and abnormal functional changes. Additionally, GDK also affects the flora related to the synthesis and metabolism of intestinal bile acids, amino acids, and short-chain fatty acids, but its specific causal relationship needs to be further explored.

## Figures and Tables

**Figure 1 antioxidants-11-02234-f001:**
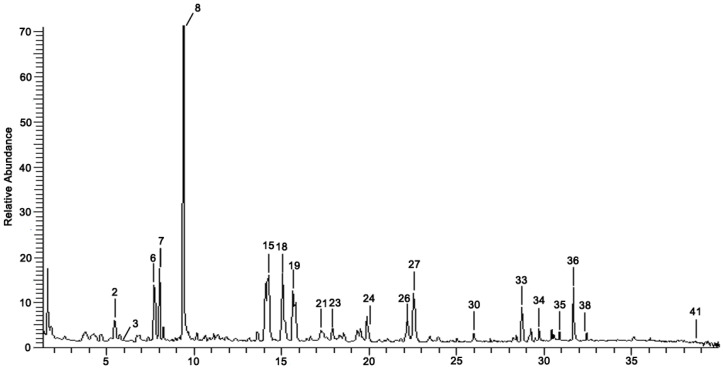
Chemical composition identification in GDK by LC-Orbitrap-ESI-MS.

**Figure 2 antioxidants-11-02234-f002:**
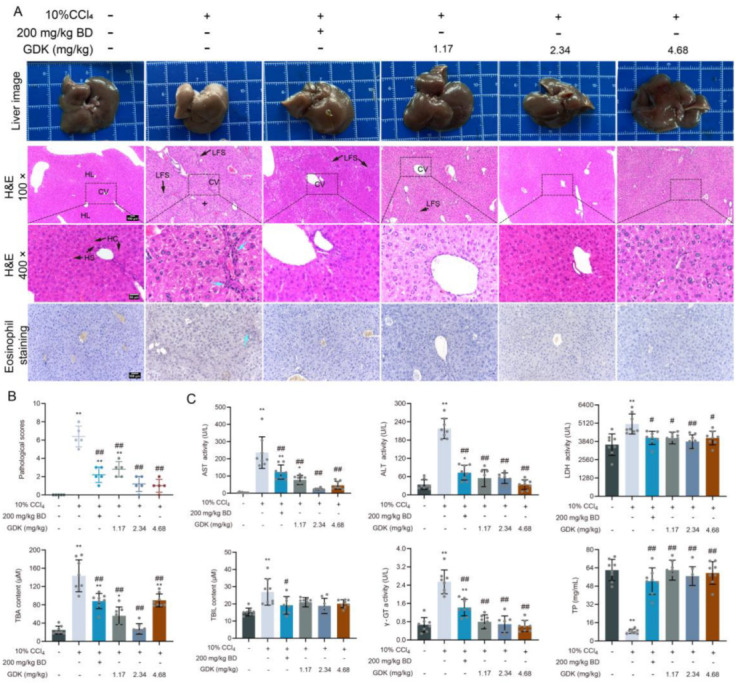
GDK improves hepatic function and structure in CCl_4_-induced chronic liver injury mice. (**A**) Representative images of the liver; hematoxylin and eosin (H&E)- and eosinophil-stained images of liver sections in indicated mice. Scar bar, 100 μm (100× and 200×), 50 μm (400×). n = 5 per group. LFS: Long fiber spacing; HL: Hepatic lobule; HS: Hepatic sinusoid; HC: Hepatic cord. Blue arrow: inflammatory cell infiltration. (**B**) Pathological scores. (**C**) Serum ALT, AST, LDH, TBIL, TBA, γ-GT, and TP levels in indicated mice. n = 8 samples per group. * *p* < 0.05, ** *p* < 0.01, # *p* < 0.05, and ## *p* < 0.01.

**Figure 3 antioxidants-11-02234-f003:**
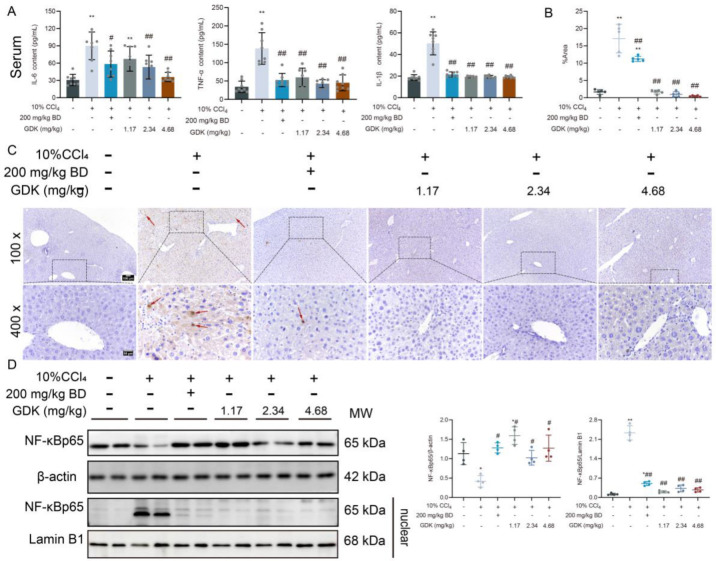
GDK reduces inflammation in CCl_4_-induced chronic liver injury mice. (**A**) Serum IL-6, TNF-α, and IL-1β levels in indicated mice (n = 8 per group). (**B**,**C**) TUNEL analysis of the liver samples of indicated mice. Red arrow: Positive area. (**D**) Western blot analysis of total and nuclear NF-κBp65 in the liver of indicated mice. Lamin B1 and β-actin served as loading controls (n = 3 per group). * *p* < 0.05, ** *p* < 0.01, # *p* < 0.05, and ## *p* < 0.01.

**Figure 4 antioxidants-11-02234-f004:**
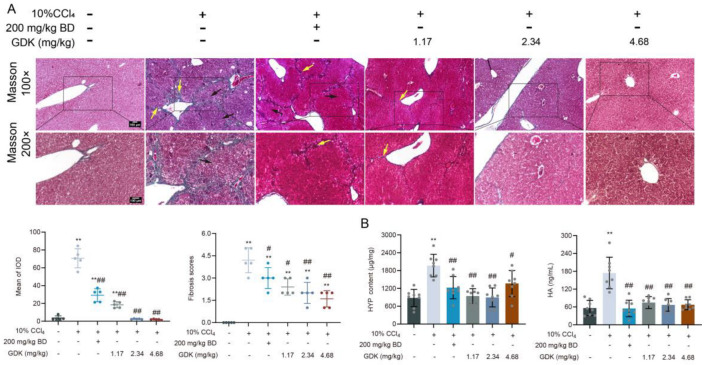
GDK suppresses fibrosis in CCl_4_-induced chronic liver injury mice. (**A**) Masson’s staining of liver sections and quantitative result in indicated mice. Scar bar, 100 μm. Black arrow: Pseudolobule; Yellow arrow: Fibrosis. (**B**) Serum HYP and liver HA levels in indicated mice (n = 8 samples per group). ** *p* < 0.01, # *p* < 0.05, and ## *p* < 0.01.

**Figure 5 antioxidants-11-02234-f005:**
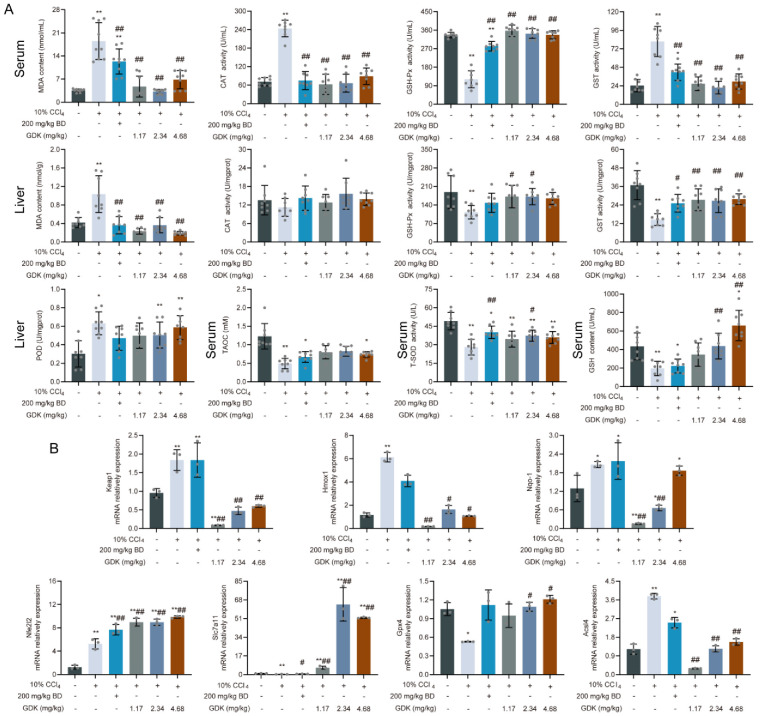
GDK attenuates ferroptosis through the antioxidant system in CCl_4_-induced chronic liver injury mice. (**A**) Levels of serum and liver antioxidants in indicated mice (n = 8 samples per group). (**B**) Relative mRNA levels of *Keap-1*, *Hmox-1*, *Nqo-1*, *Nfe2l2*, *Slc7a11*, *Gpx4*, and *Acsl4*. * *p* < 0.05, ** *p* < 0.01, # *p* < 0.05, and ## *p* < 0.01. n = 3 per group.

**Figure 6 antioxidants-11-02234-f006:**
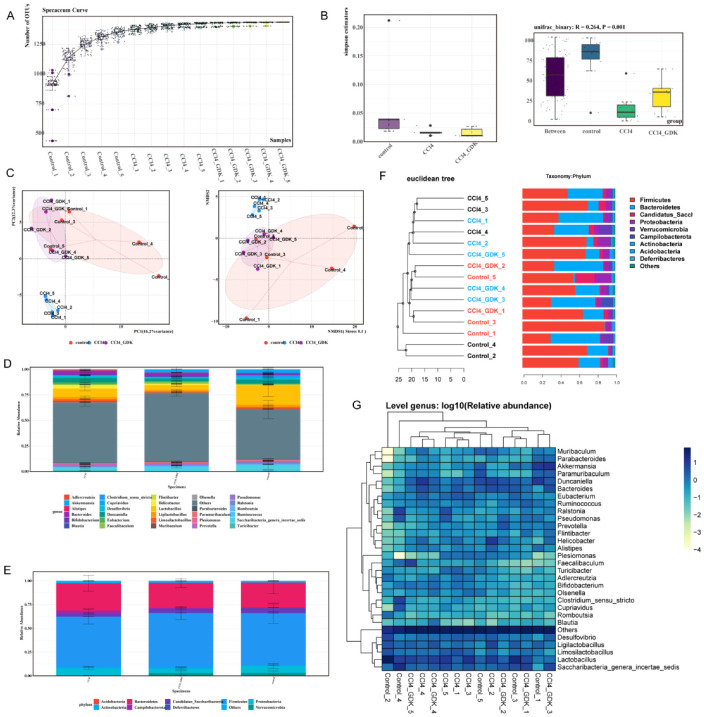
GDK restores intestinal flora diversity to curb chronic liver injury. (**A**) Species accumulation curves (**B**) α-diversity upon oral administration of GDK. (**C**) β-diversity of PCoA and NMDS plots upon GDK administration, as assessed by Euclidean distance at genus levels. (**D**,**E**) The relative abundance of bacterial phyla (**D**) and genera (**E**). (**F**) UPGMA clustering tree based on Euclidean distance and information statistics of Top 10 species at the phylum level. (**G**) Abundance cluster diagram at genus levels (n = 5 per group). R software (version 3.2.3) was used for statistics and visual display of species annotation results. It uses (https://bitbucket.org/nsegata/hclust, accessed on 23 September 2021) to construct heat maps of some species information at the OTU level and different classification levels as well as to carry out cluster analysis among samples and species.

**Figure 7 antioxidants-11-02234-f007:**
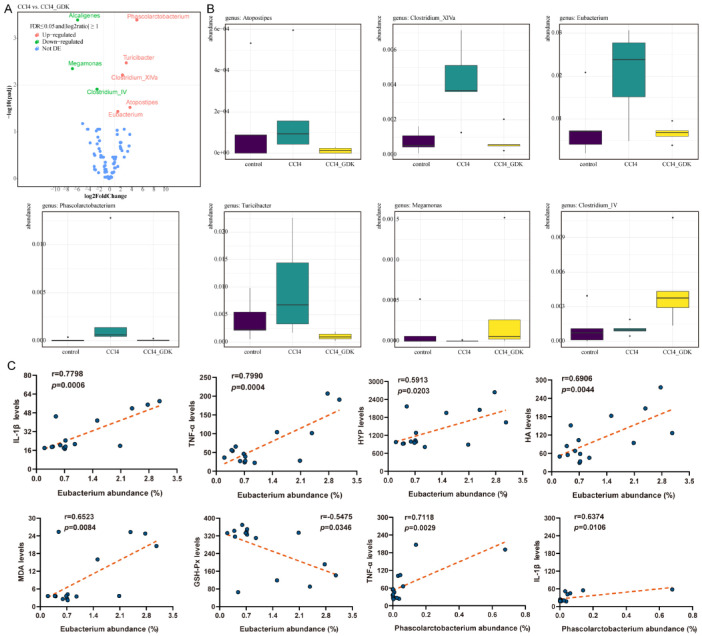
Analysis of the differences in species composition between groups upon GDK treatment. (**A**) Volcanic map of grouping difference in species composition (genus level). (**B**) Differential flora among control, CCl_4_, and GDK groups. (**C**) Correlation analysis of *Eubacterium* and *Phascolarctobacterium* with IL-1β, TNF-α, HYP, HA, γGT, MDA, GSH-Px, GST, and GSH levels based on Pearson’s Correlation Coefficient.

**Table 1 antioxidants-11-02234-t001:** Identified bioactive components of GDK with LC-Orbitrap-ESI-MS.

No	Rt (min)	Formula	Measured [M-H]−	MS/MS (*m*/*z*)	Tentative Identification	Structural Formula
1	2.69	C_17_H_24_O_9_	371.13366	191.05609	Eleutheroside B	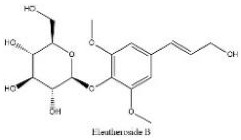
2	5.89	C_12_H_14_O_8_	285.0395	152.01164; 153.01944; 108.02183; 285.06177	Uralenneoside	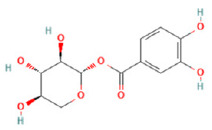
3	5.92	C_10_H_18_O_4_	201.11214	110.97005; 201.80287	Sebacic Acid	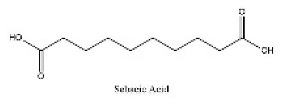
4	6.9	C_20_H_20_O_5_	339.0036	265.00217; 238.98785	Licocoumarone	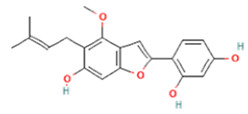
5	7.46	C_20_H_20_O_4_	323.1349	323.13495; 186.85599	licoagrochalcone A	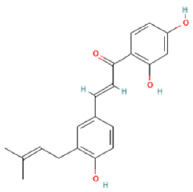
6	7.73	C_25_H_28_O_5_	407.0068	248.98572; 317.01288	6,8-Diprenylnaringenin	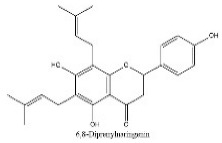
7	8.01	C_27_H_30_O_15_	593.1725	593.17535	Nicotiflorin	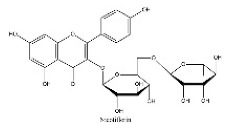
8	9.46	C_20_H_18_O_6_	353.0879	191.05623; 161.02438; 93.03436	Gancaonin C	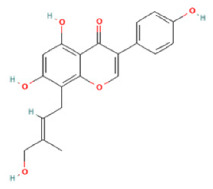
9	9.77	C_26_H_28_O_14_	563.2354	101.02442; 563.16235; 149.04539	Schaftoside	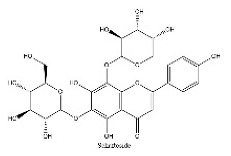
10	10.16	C_27_H_30_O_15_	593.1309	353.06696; 383.07755; 473.10950	Vicenin-2	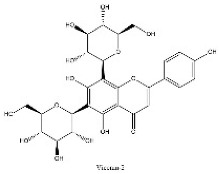
11	10.33	C_26_H_32_O_5_	423.188	221.06671; 179.05582	Licoricidin	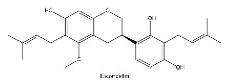
12	10.99	C_21_H_20_O_6_	367.1038	191.05617; 173.04538; 193.05045	Curcumin	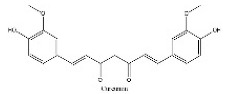
13	10.99	C_21_H_20_O_6_	367.1038	191.05617; 149.06084	Gancaonin B	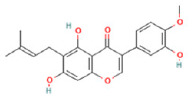
14	13.71	C_15_H_12_O_2_	223.0614	223.06140; 69.80861	Dibenzoylmethane	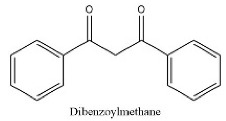
15	14.06	C_21_H_22_O_9_	417.1402	255.06669; 135.00889; 119.05022	neoliquiritin	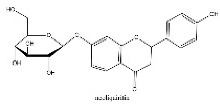
16	14.1	C_21_H_20_O_12_	463.0887	301.03516; 243.06598	Isoquercetin	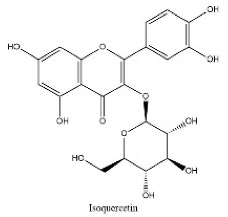
17	14.13	C_15_H_12_O_4_	255.0512	119.05028; 135.00887; 255.06685	Liquiritigenin	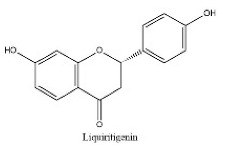
18	14.24	C_27_H_30_O_16_	609.1467	300.02774; 301.03549; 609.14716	Rutin	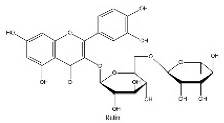
19	15.79	C_21_H_20_O_10_	431.09727	269.04592; 431.09910	Isovitexin	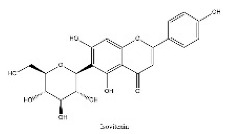
20	16.60	C_27_H_32_O_14_	579.17083	271.06131; 151.00371	naringin	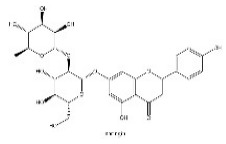
21	17.37	C_20_H_18_O_6_	353.0875	191.05620; 353.08804; 192.90018	Gancaonin O	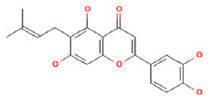
22	17.4	C_28_H_32_O_16_	623.1624	315.05094; 623.19692; 299.01936	Narcissoside	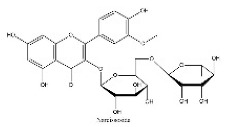
23	17.56	C_15_H_10_O_7_	301.0355	301.03574; 272.84845; 255.03066	Quercetin	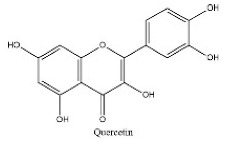
24	20.71	C_24_H_30_0_4_	381.1325	337.20242; 381.19220	Gancaonin R	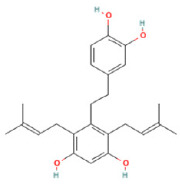
25	21.79	C_16_H_12_O_4_	267.0663	252.04292; 267.06641	Formononetin	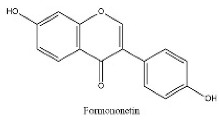
26	22.2	C_15_H_10_O_6_	285.0395	285.04072; 154.89865; 284.03243	Kaempferol	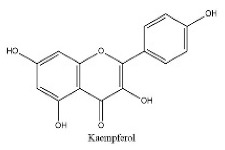
27	22.54	C_15_H_10_O_4_	253.0356	107.05025; 94.02987	Chrysophanol	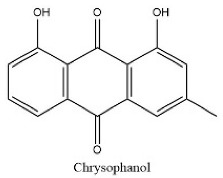
28	23.45	C_16_H_14_O_5_	285.0395	285.04056; 241.05064; 165.01947	isosakuranetin	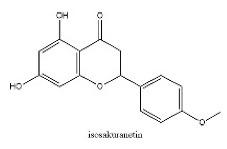
29	23.97	C_16_H_32_O_2_	255.0512	255.06648	palmitic acid	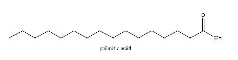
30	26.09	C_16_H_12_O_7_	315.0514	300.02780; 315.05176; 162.83954	Isorhamnetin	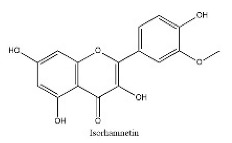
31	28.21	C_15_H_12_O_5_	271.0614	151.00372; 271.06125; 119.05020; 107.01391	naringenin	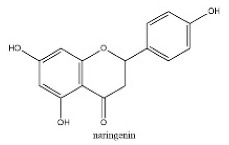
32	28.45	C_25_H_28_O_4_	391.1403	391.14026; 202.92906	(S)-4′,7-Dihydroxy-3′,8-diprenylflavanone	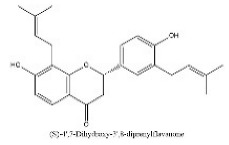
33	28.83	C_16_H_14_O_4_	269.0446	269.04580; 241.05042	Medicarpin	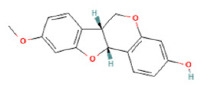
34	29.72	C_42_H_62_O_17_	837.3926	837.39126; 193.03549; 661.36050	Licorice saponin G2	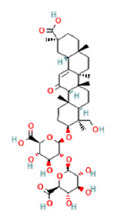
35	30.79	C_15_H_12_O_4_	255.0665	119.05023; 255.06645; 153.01938; 135.00890	Isoliquiritigenin	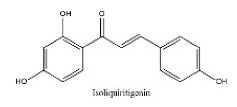
36	31.68	C_42_H_62_O_16_	821.3978	821.39753	Glycyrrhizic Acid	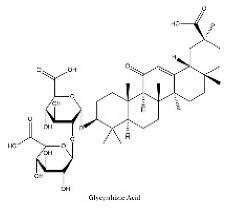
37	31.83	C_42_H_62_O_16_	821.3978	351.05722; 821.39856	Glycyrrhizin	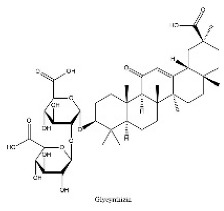
38	32.24	C_42_H_64_O_15_	807.4183	351.05713; 807.41803; 193.03543	Licorice saponin B2	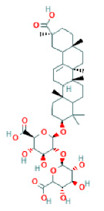
39	32.93	C_42_H_64_O_16_	823.4135	351.05748; 823.41446; 193.03565	Licorice saponin J2	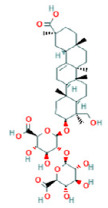
40	37.78	C_17_H_14_O_5_	297.1531	297.15314; 166.92424	3-Hydroxy-3′,4′-Dimethoxyflavone	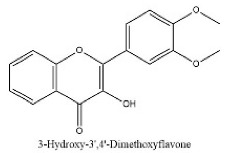
41	38.55	C_16_H_22_O_4_	277.1446	121.02955; 134.03772;	Di isobutyl phthalate	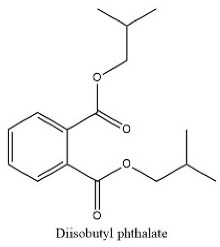

## Data Availability

Data is contained within the article and Appendix A.

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
