# Peer review of "Comprehensive Effect of Carbon Tetrachloride and Reversal of Gandankang Formula in Mice Liver: Involved in Oxidative Stress, Excessive Inflammation, and Intestinal Microflora"

_antioxidants, 2022, doi:10.3390/antiox11112234_

Round 1
Reviewer 1 Report
In the present basic science article Wei et al showed that Gandankang (a traditional Chinese drug) is able to improve an experimental model of liver fibrosis (CCl4). Additionally, Gandankang attenuated the change of gut microbiota composition induced by CCl4. Main comments:
1) Page 1 line 13: maybe Authors should add “Aim: to …”.
2) Some abbreviations should be explained (see for example page 3: ICR, SPF).
3) I feel that the discussion is too long and deserves to be shortened. On the other hand, discussion should cover the perspectives of Gandankang for human use and describe whether some studies on humans have been published.
Author Response
Reviewer 1:Comments and Suggestions for Authors
In the present basic science article Wei et al showed that Gandankang (a traditional Chinese drug) is able to improve an experimental model of liver fibrosis (CCl4). Additionally, Gandankang attenuated the change of gut microbiota composition induced by CCl4. Main comments:
1) Page 1 line 13: maybe Authors should add “Aim: to …”.
Response: We are grateful to the reviewers for their valuable comments. We added “Aim: to …” in line 13.
2) Some abbreviations should be explained (see for example page 3: ICR, SPF).
Response: Thanks. We added interpretation about this abbreviations in line 111-112, 115.
- I feel that the discussion is too long and deserves to be shortened. On the other hand, discussion should cover the perspectives of Gandankang for human use and describe whether some studies on humans have been published.
Response: We appreciate the referee for the constructive suggestions. We shortened the discussion in line 377-388, 395-398, 416-418, 425-429 and 435. And added a reference on pharmacological research of Gandankang. Since studies of Gandankang on humans really few and we only added one reference that is [47].
Reviewer 2 Report
* The authors made reasonable efforts in treating carbon tetraoxide toxicity with Chinese herbal medicine but you have recheck your manuscript by native speaker. For example: line 110, line 136.
I have some issues:
1. Line 113: the reference for CCl4 dose? I think also four weeks are too short to be called chronic. Maybe you can call it subacute. Why did you choose three times a week?
2. Line 116: Do you mean that 10% CCl4 solution was also administrated three times a week?
3. Line 131: mention the fixation duration in 4% paraformaldehyde and temperature.
4. Line 136: I think the biased measurment by analytic program such as Fiji image analysis will be better to measure the Masson trichrome, Sirius Red and immunohistochemical reactions.
5. Line 171-179: this section is not clear. where are alpha SMA, MMP1 and TMP1 in the results section? Is TUNEL technique same to immunohistochemistry?
6. I could not find Biphenyl diester in material and methods section. why did you include in the abstract and results.
7. Line 215: Sirius red is indicative to fibrosis not eosinophil infiltration. Remove it. There is no reaction.
8. Figure 2 A: there is nothing callad hyptocyte edema or eosinophilic bodies. Moreover, it seems that not all figuers in H and E 400x have the same magnification. Please, also make all figures same magnification.
9. Figure 3C: The TUNEL reaction must be in the nucleus. The reaction revealed in the figures is non specific reaction. I suggest to stain by caspase 3, ssDNA and Cox2 antibodies.
10. Figure 4A: alpha SMA staining is not specific. Please, remove it. I think all immunohistochemistry technique must be changed. All reactions are non specific and not choosed randomly.
Author Response
Reviewer 2:Comments and Suggestions for Authors
* The authors made reasonable efforts in treating carbon tetraoxide toxicity with Chinese herbal medicine but you have recheck your manuscript by native speaker. For example: line 110, line 136.
Response: We are grateful to the reviewers for their valuable comments. We have checked our manuscript by editage Editing Service again and improved the grammar through the full article. And rewrote the sentence in line 137-138. And deleted sentence in line 109-110.
I have some issues:
- Line 113: the reference for CCl4 dose? I think also four weeks are too short to be called chronic. Maybe you can call it subacute. Why did you choose three times a week?
Response: Thanks. We referred the previous study which used 10% CCl4, three times a week to established chronic liver injury model. Line 117, Ref[35, 36].
- Line 116: Do you mean that 10% CCl4 solution was also administrated three times a week?
Response: Yes. Line 117.
- Line 131: mention the fixation duration in 4% paraformaldehyde and temperature.
Response: In line 137-138, we added the fixation duration and temperature.
- Line 136: I think the biased measurment by analytic program such as Fiji image analysis will be better to measure the Masson trichrome, Sirius Red and immunohistochemical reactions.
Response: We thank the reviewer for the insights and constructive comments. So we added the description of biased measurment in line 143-145 for Masson and IHC. Quantitative result of Masson stating was added in Figure 4A. line 271-273.
- Line 171-179: this section is not clear. where are alpha SMA, MMP1 and TMP1 in the results section? Is TUNEL technique same to immunohistochemistry?
Response: Thanks, we deleted this part. Line 271-273. And revised the antibody items in Line 468.
- I could not find Biphenyl diester in material and methods section. why did you include in the abstract and results.
Response: Sorry for our negligence. Biphenyl diester (BD, 200 mg/kg) was used as positive control drug in our study which we added in line 120-121.
- Line 215: Sirius red is indicative to fibrosis not eosinophil infiltration. Remove it. There is no reaction.
Response: Thanks, we deleted it. Line 139.
- Figure 2 A: there is nothing callad hyptocyte edema or eosinophilic bodies. Moreover, it seems that not all figuers in H and E 400x have the same magnification. Please, also make all figures same magnification.
Response: We deleted the words such as hyptocyte edema or eosinophilic bodies, in line 240-241. For another question: We rechecked many times to make sure that all figuers in H and E 400x have the same magnification which could be confirmed by the view of 100x and size of nuclear. It seems different that may be due to the influence of the thickness of liver blood vessels. line 236.
- Figure 3C: The TUNEL reaction must be in the nucleus. The reaction revealed in the figures is non specific reaction. I suggest to stain by caspase 3, ssDNA and Cox2 antibodies.
Response: Thanks, we rechecked and noticed that the reaction which we marked were in the nucleus. And we also deleted some non specific mark. line 255. (Figure attached in reviewer 2.doc)
- Figure 4A: alpha SMA staining is not specific. Please, remove it. I think all immunohistochemistry technique must be changed. All reactions are non specific and not choosed randomly.
Response: Thanks, we deleted this part. Line 269, 271-273, 425-429, 468.
Finally, we sincerely hope that this revised manuscript has addressed all your comments and suggestions. We appreciated for reviewers' warm work earnestly, and hope that the correction will meet with approval. Once again, thank you very much for your comments and suggestions.

Round 2
Reviewer 1 Report
ANSWERS WERE SATISFACTORY
Author Response
Thanks for your kindly comments.
Reviewer 2 Report
* The authors made good efforts in improving their manuscript however I still have major concerns:
* You wrote you fixed your samples in 4% formaldehyde for 1 week at 25℃ which is a very long period at room temperature which will affect greatly the antigenicity of samples to be affected by antibodies.
* In relation to the TUNEL reaction, I think it is still a weak reaction and nonspecific reaction, you can check the next paper and see the difference:
Takahashi, M., N. J. Deb, Y. Kawashita, S. W. Lee, J. Furgueil, T. Okuyama, N. Roy-Chowdhury, B. Vikram, J. Roy-Chowdhury, and C. Guha. "A novel strategy for in vivo expansion of transplanted hepatocytes using preparative hepatic irradiation and FasL-induced hepatocellular apoptosis." Gene therapy 10, no. 4 (2003): 304-313.
I still suggest immunohistochemical staining by caspase 3, ssDNA, and Cox2 antibodies with good antigen retrieval for your very long fixation period of the samples.
Author Response
* The authors made good efforts in improving their manuscript however I still have major concerns:
* You wrote you fixed your samples in 4% formaldehyde for 1 week at 25℃ which is a very long period at room temperature which will affect greatly the antigenicity of samples to be affected by antibodies.
Response: We are grateful to the reviewers for their valuable comments and reminder. For the preciseness of the experiment, we deleted these results about immunohistochemistry. Line 271-273. And we will notice that in the further study that fixed our samples in 4% formaldehyde no more than 3 days at 4℃. Thanks again.
* In relation to the TUNEL reaction, I think it is still a weak reaction and nonspecific reaction, you can check the next paper and see the difference:
Takahashi, M., N. J. Deb, Y. Kawashita, S. W. Lee, J. Furgueil, T. Okuyama, N. Roy-Chowdhury, B. Vikram, J. Roy-Chowdhury, and C. Guha. "A novel strategy for in vivo expansion of transplanted hepatocytes using preparative hepatic irradiation and FasL-induced hepatocellular apoptosis." Gene therapy 10, no. 4 (2003): 304-313.
I still suggest immunohistochemical staining by caspase 3, ssDNA, and Cox2 antibodies with good antigen retrieval for your very long fixation period of the samples.
Response: Thanks! We appreciate the referee for the constructive suggestions. We also carefully selected and replaced the according images for group B and C refer to the references (Takahashi, M., N. J. Deb, Y. Kawashita, S. W. Lee, J. Furgueil, T. Okuyama, N. Roy-Chowdhury, B. Vikram, J. Roy-Chowdhury, and C. Guha. "A novel strategy for in vivo expansion of transplanted hepatocytes using preparative hepatic irradiation and FasL-induced hepatocellular apoptosis." Gene therapy 10, no. 4 (2003): 304-313.). For experiment about immunohistochemical staining by caspase 3, ssDNA, and Cox2, we intensively regret that due to the urgency of time, we failed to complete the above experiment in recent 10 days.
Figure 3. GDK reduces inflammation in CCl4-induced chronic liver injury mice. and Figure in reference (B. Vikram, J. Roy-Chowdhury, and C. Guha. "A novel strategy for in vivo expansion of transplanted hepatocytes using preparative hepatic irradiation and FasL-induced hepatocellular apoptosis." Gene therapy 10, no. 4 (2003): 304-313.) were in attachment.
Finally, we sincerely hope that this revised manuscript has addressed all your comments and suggestions. We appreciated for reviewers' warm work earnestly, and hope that the correction will meet with approval. Once again, thank you very much for your comments and suggestions.

Round 3
Reviewer 2 Report
I have no further comments and the manuscript is ready for publication